# Towards Performance-maximizing Network Pruning via Global Channel Attention

## Abstract

Network pruning has attracted increasing attention recently for its capability of significantly reducing the computational complexity of large-scale neural networks while retaining the high performance of referenced deep models. Compared to static pruning removing the same network redundancy for all samples, dynamic pruning could determine and eliminate model redundancy adaptively and obtain different sub-networks for each input that achieve state-of-the-art performance with a higher compression ratio. However, since the system has to preserve the complete network information for running-time pruning, dynamic pruning methods are usually not memory-efficient. In this paper, our interest is to explore a static alternative, dubbed GlobalPru, to conventional static pruning methods that can take into account both compression ratio and model performance maximization. Specifically, we propose a novel channel attention-based learn-to-rank algorithm to learn the optimal consistent (global) channel attention prior among all sample-specific (local) channel saliencies, based on which Bayesian-based regularization forces each sample-specific channel saliency to reach an agreement on the global channel ranking simultaneously with model training. Hence, all samples can empirically share the same pruning priority of channels to achieve channel pruning with minimal performance loss. Extensive experiments demonstrate that the proposed GlobalPru can achieve better performance than state-of-the-art static and dynamic pruning methods by significant margins.

## 1 Introduction

Convolutional neural networks (CNNs) have achieved great success in many visual recognition tasks including image classification He et al. (2016), object detection Ren et al. (2015), image segmentation Dai et al. (2016), etc. The success of CNNs is inseparable from an excessive number of parameters that are well organized to perform sophisticated computations, which conflicts with the increasing demand for deploying these resource-consuming applications on resource-limited devices.

Network pruning has been proposed to effectively reduce the deep model's resource cost without a significant accuracy drop. Unstructured pruning methods Han et al. (2015b;a) usually reach a higher compression rate, while relying on dedicated hardwares/libraries to achieve the actual effect. In contrast, structured pruning methods Li et al. (2016); He et al. (2017b) preserve the original convolutional structure and are more hardware-friendly. Considering the greater reduction in terms of floating-point operations (FLOPs) and hardware commonality, this research focuses on channel pruning. Existing methods perform channel pruning either statically or dynamically. Static pruning methods remove the same channels for all images Molchanov et al. (2019); Tang et al. (2020) while dynamic pruning removes different channels for different images Rao et al. (2018); Tang et al. (2021b). However, both existing static methods and dynamic methods have some limitations. Particularly, given the fact that the channel redundancy is highly sample-dependent, static pruning methods may remove some channels that are not redundant for certain images. Consequently, static methods refrain from a larger pruning rate to avoid a significant accuracy drop. To tackle the issue of image-specific redundant channels, dynamic pruning methods remove image-specific channels. In this way, dynamic methods achieve the state-of-the-art pruning ratio without significantly sacrificing performance. Despite the significant advantage, dynamic pruning usually requires preserving the full original model during inference, which restricts its practical deployment on resource-limited devices.

In this paper, we propose a new paradigm of static pruning method named GlobalPru. Although born from static, GlobalPru tackles the issue of image-specific redundant channels by making all images share the same ranking of channels with respect to redundancy. In other words, GlobalPru forces all images to agree on the same ranking of channel saliency (referred to as global channel ranking) to reduce the image-specific channel redundancy to the greatest extent possible. By removing channels with the lowest global rankings, GlobalPru can avoid the problem of existing static pruning methods, which remove more important channels while retaining less important channels for specific images with a high probability. More specifically, we first propose a novel global channel attention mechanism. Channel attention is local in the sense that the ranking of channels with respect to their importance is image-specific. Different from existing ones, our proposed global channel attention mechanism can identify the global channel ranking across all different samples in the training set, particularly through a learn-to-rank regularization. In detail, to make the static GlobalPru approach the maximum image-specific compression ratio of dynamic pruning as well as stabilize the training process, we first use a majority-voting-based strategy to specify the global ranking to make the static GlobalPru approach the maximum image-specific compression ratio of dynamic pruning. Then, given a certain ranking, all the image-specific channel rankings are forced to agree on the ranking via learn-to-rank regularization. When all the image-specific channel rankings are the same as the given ranking, the ranking becomes the global ranking. As a result of exposing global ranking for all images during training stages, GlobalPru can also avoid the disadvantage of existing dynamic pruning which needs to store the entire model for deciding image-specific channel ranking during inference and perform more efficient pruning on globally ordered channels.

Our contributions are summarized as follows:

- We propose GloablPru, a static network pruning method. GlobalPru tackles the issue of image-specific channel redundancy faced by existing static methods by learning a global ranking of channels w.r.t redundancy. GlobalPru produces a pruned network such that GlobalPru is a more memory-efficient solution than existing dynamic methods.

- To the best of our knowledge, we for the first time propose a global channel attention mechanism where all the images share the same ranking of channels w.r.t. importance. Instead of repeatedly computing image-specific channel rankings under existing local attention mechanisms, our proposed global attention enriches the representation capacity of models and therefore greatly improves the pruning efficiency.

- Extensive experimental results show that GlobalPru can achieve state-of-the-art performance with almost all popular convolution neural network architectures.

## 2 RELATED WORK

### 2.1 STATIC PRUNING & DYNAMIC PRUNING

As the most traditional and classic model pruning method, static pruning shares a compact model among all different samples Wen et al. (2016); Liu et al. (2017a); Liebenwein et al. (2019); Molchanov et al. (2019); Tang et al. (2020). To be specific, static methods select pruning results through trade-offs on different samples, which leads to final compact models having limited representation capacity and thus suffering an obvious accuracy drop with large pruning rates. Recently, some works turn their attention to the pursuit of the ultimate pruning rate and focus on excavating sample-wise model redundancy, named dynamic pruning. Dynamic pruning generates different compact models for different samples Dong et al. (2017); Gao et al. (2018); Hua et al. (2019); Rao et al. (2018); Tang et al. (2021b). Actually, dynamic methods learn a path-decision module to find the optimal model path for each input during inference. For example, state-of-the-art work Liu et al. (2019) investigates a feature decay regularization to identify informative features for different samples, and therefore achieves an intermediate feature map to the model sparsity. Tang et al. (2021a) further improves the dynamic pruning efficiency by embedding the manifold information of all samples into the space of pruned networks. Despite dynamic methods achieve higher compression rate, most of them are not memory-efficient cause most of them requires to deploy the full model even in inference stage.

## 2.2 CHANNEL ATTENTION

Channel attention is similar to the scale coefficients added on feature channels to enhance the important features and also weaken the unimportant ones. It is usually implemented through extra channel attention modules, which can exploit the inter-channel relationship for input feature maps. An typical channel attention mechanism is SENet Hu et al. (2017), it uses the "Squeeze-and-Excitation" (SE) block which can be stacked together to adaptively recalibrate channel-wise feature responses on each convolutional layer by explicitly modelling inter-dependencies between channels. Li et al. (2019) proposes a dynamic selection mechanism in CNNs that allows each neuron to adaptively adjust its receptive field size based on multiple scales of input information, which is named "Selective Kernel" (SK) unit. By dynamically calculating channel attention on the different kernels, SK realizes parameter sharing and thus improves the model efficiency. In summary, nearly all the existing channel attention are local in the sense that the channel saliency is image-specific and can not identify a global channel attention over the entire dataset.

## 3 GLOBAL ATTENTION-BASED CHANNEL PRUNING

In this section, we will give a detailed formulation and theoretical explanation of the proposed Global Channel Attention Pruning (GlobalPru). As illustrated in Section 3.1, the GlobalPru is formulated as a static alternative regularized simultaneously by local channel-wise saliency and the global channel rank loss and can be optimized in an end-to-end manner. Then, the training process of GlobalPru is divided into two stages including Global Channel Attention Election (stage 1) and Learn-to rank Regularization Pruning (stage 2), which would be illustrated and proven theoretically in Section 3.2 and Section 3.3, respectively. In stage 1, GlobalPru explicitly elects the global channel attention that benefits most samples from the observations of the locally image-specific channel dependencies. After that, in stage 2, the proposed learn-to-rank algorithm forces the channels to be ordered towards the global channel attention concurrently during the model training process.

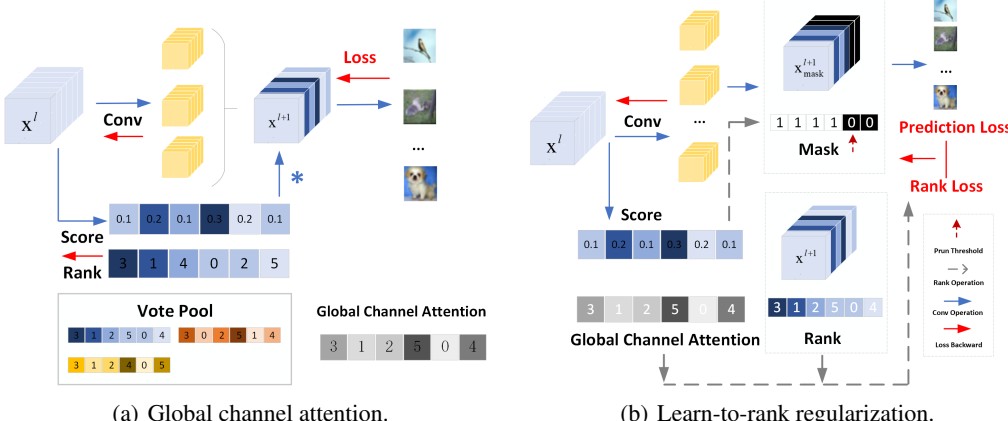

(a) Global channel attention.      (b) Learn-to-rank regularization.

Figure 1: The overview of the training process of Channel Rank Network.

## 3.1 PROBLEM FORMULATION

Given a dataset with $N$ samples as $X = \{x_i\}_{i=1}^{N}$ with the corresponding labels $Y = \{y_i\}_{i=1}^{N}$. For a convolution neural network (CNN) with $L$ convolution layers and parameters set $\Theta$, $W^l \in R^{c^l \times c^{l-1} \times k^l \times k^l}$ denotes the convolution parameters in the $l$-th layer, where $c^l$ is the channel numbers of $l$-th layer and $k^l$ represents the corresponding kernel size. $F^l(x_i) \in R^{b \times c^l \times h \times w}$ is the output feature map of the $l$-th layer for the input $x_i$, where $b$, $h$, $w$ are the batch size, height and width of the output feature map, respectively. Given the input feature $F^{l-1}(x_i)$, the output of layer $l$ can be calculated as $F^l(x_i) = Relu(Bn(W^l \otimes F^{l-1}))$, where $Relu$ and $Bn$ represent the activation and

batch normalization operation respectively. Finally, we use $f(X, \Theta)$ to represent the output of the neural network over input samples $X$.

Static channel pruning eliminates the same redundancies and discovers a compact network among all images. And most static methods are guided by empirical risk minimization and numeric-based regular terms, i.e., parameter magnitudes, channel saliencies or reconstruction errors, etc. Typical static methods can be formulated as follows:

$$\min_{\Theta} \sum_{i=1}^{N} \mathscr{L}\left(f(x_i, \Theta), y_i\right) + \lambda \cdot Norm(\Theta), \tag{1}$$

where $\mathscr{L}$ denotes the loss function and $Norm(\cdot)$ is the regularization term which is usually a human-designed criterion for inducing model sparsity. And $\lambda$ is used as a knob to strike the different trade-offs between model accuracy and sparsity ratio.

Conversely, dynamic channel pruning discovers effective sub-networks for each input dependently during the inference stage, which is usually implemented through additional model path-finding functions. Define a scoring module $S$ (a squeeze and excitation module Hu et al. (2017) in our work), for a specific input $x_i$, the channel saliencies on layer $l$ can be computed as $\pi^l(x_i) = S^l(x_i) \in R^{c^l}$, where $\pi_j^l(x_i)$ indicates the relative importance score of the $j$-th channel, and a smaller $\pi_j^l$ indicates that channel $j$ is less important. Given a pruning threshold $\epsilon^l$, the variable $mask_j^l$ to indicate the removal of the $j$-th channel of layer $l$ is initialized to 1 and set to 0 when $\pi_j^l(x_i) < \epsilon^l$. The channel saliencies for the pruned model would be re-calculated as:

$$\hat{\pi}^l(x_i) = \pi^l(x_i) \otimes mask^l(x_i) \tag{2}$$

And the feature output of the $l$-th layer would be:

$$F^l(x_i) = Relu\left(BN\left(\left(mask^l(x_i) \cdot \hat{F}^{l-1}(x_i)\right) \otimes W^l\right)\right) \tag{3}$$

Thus, a general dynamic pruning paradigm could be formulated as EQ. 4, wherein the regularization is used to induce the instant-wise network sparsity.

$$\min_{\Theta} \sum_{i=1}^{N} \mathscr{L}\left(f(x_i, \Theta), y_i\right) + \lambda \sum_{l=1}^{L} ||\pi^l(x_i)|| \tag{4}$$

Dynamic pruning has achieved higher compression rates than static methods by removing instance-specific model redundancy, however, the calculation of $\pi^l(x_i)$ depends on the complete referenced model along with additional path-addressing units being deployed on resource-limited devices. These unavoidable dependencies make dynamic methods not memory and computation-efficient, thus without great applicability.

To utilize advantages from both static and dynamic methods while avoiding the corresponding bottlenecks, GlobalPru expects all these samples to reach an agreement on the channel importance ranking simultaneously with model training, thus achieving a dynamic-like static channel pruning. To this end, GlobalPru first investigates the local channel attention of different samples via a preheated scoring module $S$, based on which it could identify a global channel attention as the channel importance ranking prior through the **Majority Vote Mechanism**. After that, GlobalPru forces all image-specific channel rankings to converge to the certain global channel ranking by the novel **Learn-to-Rank** regularization. The detailed formulation of GlobalPru is as follows.

Let $R^l$ denote the medium results of channel importance ranking of layer $l$ most voted by all input samples during the training process, and $T^l$ is the corresponding channel ranking prior. Hence he channel ranking loss could be computed as $\Phi(R_l, T_l)$, wherein $\Phi$ is the learn-to-rank regularization defined in Eq. 6,7. The optimization target of GlobalPru can be formulated as:

$$\min_{\Theta} \sum_{i=1}^{N} \sum_{l=1}^{L} \mathscr{L}(f(x_i, \Theta), y_i) + \alpha \left\|\pi^l(x_i)\right\|_1 + \beta \sum_{l=1}^{L} \Phi(R^l, T^l), \tag{5}$$

Let $T_j^l$ and $R_j^l$ represent the importance ranking index of the $j$-th channel in the prior and the global attention during training, respectively. We elaborately use the previously defined channel score $|\pi_j^l|$ to measure the similarity between the current position and the target position of the $j$-th channel on

layer $l$, and have $P(R_j^l = T_j^l) = \frac{exp(\pi_j^l)}{\sum_{j=1}^{c^l} exp(\pi_j^l)}$. Hence $\Phi$ can be formulated in Bayesian perspective as:

$$\Phi(T^l, R^l) = -\log \prod_{j=1}^{c^l} P(R_j^l = T_j^l | T_j^l) = -\sum_{j=1}^{c^l} \log \frac{P(T_j^l | R_j^l = T_j^l) P(R_j^l = T_j^l)}{P(T_j^l)} \qquad (6)$$

Since $T^l$ is defined as the prior of the channel rank and independent of the current channel ranking, we have $P(T^l) = 1$ and $P(T^l | R^l = T^l) = 1$, hence we can rewrite EQ. 6 as:

$$\Phi(T^l, R^l) = -\sum_{j=1}^{c^l} \log P(R_j^l = T_j^l) = --\sum_{j=1}^{c^l} \log \frac{exp(\pi_j^l)}{\sum_{j=1}^{c^l} exp(\pi_j^l)} \qquad (7)$$

To simulate the impact of model pruning and reduce computational overhead, we use $\hat{\pi}_j^l$ instead of $\pi_j^l$ in actual optimization.

## 3.2 GLOBAL CHANNEL ATTENTION

**Definition 1** *Channel local attention refers to the channel attention of a single input (channel importance ranking), while channel global attention refers to the channel attention consistent with the most samples for the current data domain.*

Lots of previous works have proven that the importance ranking of channels Tang et al. (2021b) is highly sample-dependent, which means the model redundancy for different samples could be quite different with high probability. In fact, natural images are a mixture of some intrinsic features such as color, edges, textures, etc., in different ratios, resulting in different requirements of different samples for the model. Channel attention actually adjusts the weight of each channel of the output features for each sample to approximate the target feature mapping. Thus, we give the following corollary.

**Corollary 1** *Conventional channel attention could indicate the quantitative channel redundancy for each sample through the learnable feature scaling factors.*

**Corollary 2** *Local channel attention is highly sample-related, which implies that redundant channels are disordered and unpredictable for the whole sample domain.*

To implement global channel attention across samples and therefore eliminate the dependency of dynamic pruning on sample-wise path-finding, we propose to explore the "Global Channel Attention" for all different samples. As shown in Figure 1(a), the global channel attention is identified by the majority vote mechanism. Specifically, a deep neural network with channel attention modules is first pre-warmed. After that, in the training stage, every image-specific local channel attention is collected as the electoral college to be voted for global channel attention, which would be the prior of the channel importance ranking in the learn-to-rank regularization. In this way, as a static alternative, GlobalPru obtains the most supported channel importance ordering in the current data domain to achieve high-quality dynamic-like pruning, achieving a higher model compression rate and reducing the precision decline from pruning.

## 3.3 LEARN-TO-RANK REGULARIZATION

The learn-to-rank regularization is explicitly modeled from Bayesian perspective as a joint probability maximization problem. As shown in Figure 1(b) and EQ. 8, specifically, regarding the global channel attention as the prior of the channel importance ranking, the channel ranking loss can be formalized as the negative value of the probability that the current channel ranking is equal to the given prior. Thus, in addition to empirical risk minimization, GlobalPru trains a model distribution towards the probabilistic model which is most likely to generate the target channel ranking. However, the huge number of channels in CNNs brings unique challenges for computing the ranking loss for each channel. We use the unique advantages of model pruning to address the problem of huge computational costs in EQ. 8, given the pruning rate $p$, the computation cost could be further reduced by maximizing the probability that only the first $(1 - p)c^l$ channels in $T^l$ are ranked correctly. The

proposed learn-to-rank regularization could be reformatted to Eq. 8, and:

$$\Phi(T^l, R^l) = -\sum_{j=1}^{(1-p)c^l} \left\{ \log exp(\pi_j^l) - \log \sum_{j=1}^{(1-p)c^l} exp(\pi_j^l) \right\} \qquad (8)$$

## 4 EXPERIMENT

In this section, GlobalPru is empirically tested on the most representative neural network architectures: plain architecture, residual structure, and lightweight depthwise convolution networks to verify the effectiveness of the proposed "Global Attention Mechanism" and "Learn-To-Rank" method.

**Datasets**: The effectiveness of the proposed GlobalPru is verified on most popular image datasets of model pruning, including MiniImageNet, CIFAR-10, CIFAR100 and SVHN. Diverse datasets can help effectively demonstrate the broad applicability of our GlobalPru.

**Models**: We explore the effectiveness of GlobalPru on three mainstream neural network architectures includes: plain architecture - VGG-16, residual structure - ResNet with different depth, and lightweight depthwise convolution MobileNet-V2. These three representative architectures cover almost all the most popular convolutional networks currently.

**Tasks**: Most model pruning works only demonstrate the accuracy of the pruned models on the image classification task, and ignores the robustness of the pruned networks. To further demonstrate the applicability of our method, we also evaluate the performance of the pruned network against adversarial perturbations on the adversarial samples detection task.

**Implementation Details**: Standard data argumentation RandomSizedCrop and RandomHorizontalFlip are used in all datasets we used. The coefficient $\alpha$ to regulate the channel saliency score is set as 0.0001 in our work. And another coefficient $\beta$ to balance the weight of the channel rank loss is also set as 0.0001 empirically. The stochastic gradient descent (SGD) is used for all training processes. Two pruning modes are tested in GlobalPru, named "Fixed" and "Mixed", respectively. Under the fixed mode, all layers use the same pruning rate with the Lottery Ticket of model pruning which corresponds to the highest test accuracy Frankle & Carbin (2019). For ResNet-18 on CIFAR10 in this work, the range of pruning rates which can achieve higher test accuracy than the original model is [38.9%, 100.0%], and the optimal pruning rate corresponds to the highest test accuracy is 61.1% Under the mixed mode, the layer-wise pruning rates are set according to the sensitivity of each layer to model perturbation. Dong et al. (2020) has theoretically proved that the right sensitivity metric for model perturbation is the average Hessian trace. Align the value of the average Hessian trace of all layers with the optimal pruning rate, and the fluctuation of the pruning rate of each layer relative to the optimal pruning rate is set according to the standard deviation of the Hessian trace of this layer relative to the model average . The larger the Hessian trace value, the lower the pruning rate. All of the experiments are conducted on NVIDIA GeForce GPUs.

### 4.1 COMPARISON ON PLAIN ARCHITECTURE

The proposed method is first compared with the state-of-the-art network pruning algorithm on the most intuitive plain architecture, which is the initially successful way to construct a convolutional neural network. The typical plain network VGG-16 we used consists of a few building blocks (convolutional layers, activation layers, etc.) stacked in a vertical manner. A "Squeeze-and-Excitation" module is added following the convolutional layer to measure its relative channel saliency. Both static methods including Luo et al. (2017); Li et al. (2017); He et al. (2017a); Liu et al. (2017b), and the dynamic methods including Lin et al. (2017); Gao et al. (2019) are compared. All of the algorithms are tested on the Cifar-10 dataset and the comparison results are shown in Table 1. The pruning performance is reported over three main metrics: Memory Access Cost (MAC), FLoating point Operations (FLOPs), and Classification Accuracy Drop. When using the fixed pruning rate for all the computing layers, our method achieves 55.2% FLOPs decrease with a negligible accuracy drop of −0.21%. And when using the empirically mixed pruning rate for each layer, our method can prune more 1.5% FLOPs than the fixed mode, while with less accuracy degradation. The results show that our approach achieves the highest compression ratio on VGG-16 with competitive model performance. Compared to the SOTA static pruning method, undoubtedly, our method wins an

Table 1: Comparison of pruning performance on VGG-16 architecture and Cifar-10 dataset. Our method is compared with different channel pruning algorithms including both traditional static methods and sample-wise dynamic methods. "Dy?" means if the current method is a sample-wise dynamic method. "Fixed" and "Mixed" represent the above-mentioned fixed pruning mode and mixed pruning mode, respectively. "*" denotes the new static pruning paradigm, which approximates the maximum compression ratio of dynamic methods.

| Method | Cite | Dy? | ↓MAC | FLOPs(M) | ↓FLOPs | -Acc |
|---|---|---|---|---|---|---|
| Baseline | | | 0 | 626.4 | 0 | 0 |
| LIWS | Wang et al. (2020) | Y | 53.2% | 184 | 70.6% | -0.8% |
| ThiNet | Luo et al. (2017) | N | 50% | 313 | 50% | -0.14% |
| L1-norm | Li et al. (2017) | N | 34% | 413.2 | 34% | -0.5% |
| CP | He et al. (2017a) | N | 50% | 313 | 50% | -0.32% |
| NS | Liu et al. (2017b) | N | 51% | 306.8 | 51% | -0.19% |
| RNP | Lin et al. (2017) | Y | 50% | 313 | 50% | -0.85% |
| FBS | Gao et al. (2019) | Y | 50% | 310.4 | 50.4% | -0.47% |
| GlobalPru(Fixed) | Ours | * | 55% | 280.87 | 55.2% | -0.21% |
| GlobalPru(Mixed) | Ours | * | 56.7% | 271.26 | 56.7% | -0.15% |

Table 2: Comparison of pruning performance on Residual architecture: ResNet-20 architecture and Cifar-10 dataset.

| Method | Cite | Dy? | FLOPs(M) | ↓FLOPs | -Acc |
|---|---|---|---|---|---|
| Baseline | | | 43.17 | 0 | 0 |
| SFP | He et al. (2018) | N | 24.95 | 42.2% | -1.39% |
| FPGM | He et al. (2019) | N | 19.86 | 54% | -1.78% |
| DSA | Ning et al. (2020) | N | 21.46 | 50.3% | -0.84% |
| Hinge | Li et al. (2020a) | N | 23.53 | 45.5% | -0.38% |
| DHP | Li et al. (2020b) | N | 20.80 | 51.8% | -0.68% |
| Maninp | Tang et al. (2021a) | Y | 19.77 | 54.2% | -0.17% |
| FBS | Gao et al. (2019) | Y | 19.97 | 53.1% | -1.25% |
| GlobalPru(Fixed) | Ours | * | 15.77 | 60.84% | -0.14% |
| GlobalPru(Mixed) | Ours | * | 16.07 | 60.1% | -0.26% |

absolute advantage in pruning rate and only has a slight accuracy drop, i.e., $-0.07\%$ and $-0.01$ respectively for fixed and mixed modes, than state-of-the-art ThiNet. And when compared to the pioneering dynamic pruning method, our method gains and obviously less accuracy loss than the newest Liu et al. (2019), and outperforms Lin et al. (2017); Gao et al. (2019) in both compression rate and accuracy.

## 4.2 COMPARISON ON RESIDUAL ARCHITECTURE

We further explore the effectiveness of GlobalPru on the residual neural networks, where GlobalPru explores global attention inside residual blocks by adding the vanilla input feature representations to the output feature maps scaled by channel attention. To be specific, we introduce compression on each convolutional layer inside the residual block without changing the transfer of residual information. The compared methods include both SOTA static method He et al. (2018; 2019); Ning et al. (2020); Li

Table 3: Comparison of pruning performance on DepthWise architecture: MobileNet-V2 architecture and Cifar-10 dataset.

| Method | Cite | Dy? | FLOPs(M) | ↓FLOPs | -Acc |
|---|---|---|---|---|---|
| Baseline | | | 89.9 | 0 | 0 |
| WM | Zhuang et al. (2018) | N | 66.53 | 26% | -3.68% |
| DCP | Zhuang et al. (2018) | N | 66.53 | 26% | -3.16% |
| NPPM | Gao et al. (2021) | N | 47.65 | 47% | -3.1% |
| GlobalPru(Fixed) | Ours | * | 47.47 | 47.2% | -0.69% |
| GlobalPru(Fixed) | Ours | * | 35.83 | 60.14% | -0.55% |

et al. (2020a;b) and state-of-the-art dynamic pruning method Tang et al. (2021a) and Gao et al. (2019). The results are tested on ResNet-20 and Cifar-10. As shown in Table 2, the FLOPs, FLOPs decrease relative to the baseline and post pruning accuracy drop are reported. Our method under both fixed pruning rate and mixed pruning rate setting achieve the highest compression ratio at the same time with the most negligible accuracy drop. For example, even under the highest pruning rate 68% for all layers, the accuracy of our method still reaches 98.01% with only −0.34 degradation. Besides, we also compare the performance of GlobalPru with the classic three-stage "train-prune-fine-tune" static pruning at different pruning rates. According to Figure **??**, we choose 40.1% and 61.1% as model pruning rates. They correspond to the point where the generalization performance is the highest and the point where the generalization accuracy starts to suffer, respectively. All of the comparisons show that GlobalPru can work efficiently on residual neural networks.

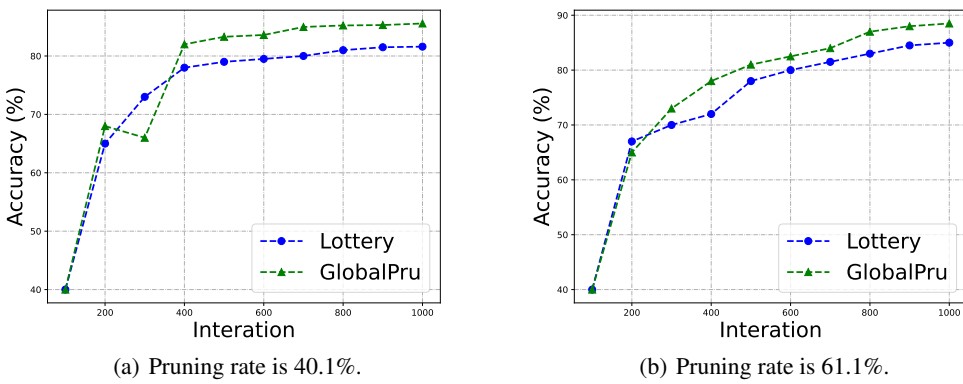

(a) Pruning rate is 40.1%.          (b) Pruning rate is 61.1%.

Figure 2: Comparison of pruning performance under different pruning rates on ResNet-20 and Cifar-10. "Lottery" represents an typical three-stage static pruning paradigm.

## 4.3 COMPARISON ON DEPTHWISE ARCHITECTURE

Another popular model architecture using "Depthwise Separable Convolution" is also included in our experiment to cover another class of models in the spotlight. Also, the depthwise architecture is originally designed for resource-limited mobile devices and is already very light-weight. To validate whether our method can further compress these already compressed models, we compare it with the most widely used efficient MobileNet-V2. As shown in Table 3, while outperforming all methods in computing reduction, our model achieves the closest model accuracy to the baseline. The advantage of our method on lightweight DepthWise architecture is obvious which proves that our model enables further model compression in an almost lossless mode over current state-of-the-art methods.

Table 4: Comparison of pruning performance on MiniImageNet (M-ImageNet) and SVHN datasets with popular model pruning SOTAs.

| Dataset | Method | Accuracy (%) | | ↓Size |
|---|---|---|---|---|
| | | Top 1 | Top 5 | |
| M-ImageNet | PolarPrun | 85.857 | 96.068 | 0.59 |
| | SlimPrun | 70.06 | 90.44 | 0.60 |
| | **GlobalPru** | **89.227** | **96.97** | **0.60** |
| SVHN | PolarPrun | 97.099 | 98.199 | 0.59 |
| | SlimPrun | 96.848 | 99.228 | 0.50 |
| | **GlobalPru** | **97.543** | **99.3** | **0.60** |

Table 5: Comparison of AUROC (%) of GlobalPru with typical adversarial detection methods on Fast Gradient Sign Method (FGSM)-based CIFAR100 and SVHN datasets .

| AUROC (%) FGSM | Non Prune | | Prune | |
|---|---|---|---|---|
| | MAHA | FBS | DPIC | **GlobalPru** |
| SVHN | 99.63 | 99.95 | 99.96 | **100.00** |
| CIFAR100 | 99.77 | 100.00 | 100.00 | **100.00** |

## 4.4 GENERALIZATION VERIFICATION

In addition to model accuracy, a good pruning method also needs to have good generalization and robustness, which is usually ignored in previous work. To show that GlobalPru has these favorable properties, we test the performance of GlobalPru on a more diverse vision dataset and extra visual tasks with added adversarial samples.

As shown in Table.4, we compare the performance of GlobalPru and two static pruning SOTAs on typical image datasets SVHN and MiniImageNet. SVHN contains large-scaled labeled data (over 600,000 digit images) for a more difficult, unsolved real-world problem. MiniImageNet, drawn from ImageNet, has complex samples and multi-classes, more suitable for prototyping and experimental research. Obviously, as a static alternative, GlobalPru achieves a higher model compression/task loss ratio than static pruning SOTA by removing the global redundancy of the model for all samples.

As shown in Table.5, we verify the robustness of our method by observing the performance of GlobalPru when faced with adversarial perturbations. It can be seen that GlobalPru exhibits extraordinary perturbation resistance, even surpassing some typical adversarial sample detection methods. We speculate that this is because the model redundancy removed by GlobalPru is selected by a majority vote of all samples, thereby weakening the effect of small image disturbances and making GlobalPru more robust than the previous pruning criterion only guided by task loss.

## 5 CONCLUSION

To sum up, we propose a novel pruning method, i.e., the Channel Attention-based Learn-to-Rank Network, based on the shortcomings of current dynamic pruning methods, such as the need to save the complete model locally and repeat the forward computation. Our approach first explores the channel saliency rank of each sample, and then selects the most suitable channel rank supported by all current inputs through a majority voting mechanism. We define this channel rank as global channel attention. Next, we obtain a Channel Rank Network by proposing an efficient channel sorting algorithm to incorporate the knowledge of global channel attention into the training of the model, so as to quickly give an appropriate pruning response when new samples or sparse requirements arrive. While obtaining the advantages of dynamic pruning, our method avoids the defects of the original dynamic pruning method, and achieves better pruning performance than most state-of-the-art methods.

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
