# OpenReview forum: "Towards Performance-maximizing Network Pruning via Global Channel Attention"
_ICLR.cc/2023/Conference — Submitted to ICLR 2023_

### Official Review · Reviewer_ywfs · 2022-10-17

**Confidence:** 5
**Correctness:** 2
**Technical Novelty And Significance:** 2
**Empirical Novelty And Significance:** 2
**Recommendation:** 3

**Clarity, Quality, Novelty And Reproducibility:**

This paper is a very rushing paper with a lot of typos, unclear and confusing points.
I do not think this is a proper manner for reviewers for a top conference submissions.
See weakness points for more details.


**Strength And Weaknesses:**

Strong points
+  The bayesian based formulation for learning-to-rank is somewhat novel to me.


Weak Points
- This is a very rushing paper with a lot of typos and errors.

       - There are plenty of repeated references. i.e., many identical references are listed more than once in the reference section.
       - page-8, there are missing ref for Figure ??
       - Eq-7, there are two minus after =.
       - learn-to-rank should be "learning-to-rank" generally.

- Some descriptions are not complete, for instance
      - what are the network architecture for M-ImageNet and SVHN in Table-4? And what is the original performance?

- Some confused points:
      - in the contribution claims, "GlobalPru" is a static pruning method. However, in Table-1/2, you marked it as *, which means it is both static or dynamic, right? And in eq-5, you include the \pi(x) term which is dynamic, right? If it is static, you should exclude this term.
If not, the contribution of those two terms need investigated.

     - It claims "global". From description, it is still pruned layer-by-layer? especially for the "fixed" scenario. For the "mixed" case, it is unclear how it was realized. Just according to the ranking of $\pi_j$ from all layers?

- It lacks experiments on large-scale network on large-scale dataset like ImageNet, which this is done for most of the compared citations in the paper.

- page-5, for the corollary-1/2, do you have any mathematical proof or deductive process? Otherwise, they are just some of your claims.

**Summary Of The Paper:**

This paper proposes a channel pruning method which aims to enhance global consistence among samples for network channels.
It claims integrating both static and dynamic information into the formulation.
Comparison on small datasets demonstrate the effectiveness of the proposed methods.

**Summary Of The Review:**

See previous comments.

---

### Official Review · Reviewer_3k5Z · 2022-10-23

**Confidence:** 3
**Correctness:** 3
**Technical Novelty And Significance:** 2
**Empirical Novelty And Significance:** 2
**Recommendation:** 3

**Clarity, Quality, Novelty And Reproducibility:**

Quality: fair
Clarity: good
Novelty : fair
Reproducibility: fair


**Strength And Weaknesses:**

Weaknesses
1. The major concern is that the experiment comparison. I find the paper does not compare with sufficient recent works. Thus I may doubt why the performance is SOTA or not.
2. There is no experiments on ImageNet.
3. Experiments are too weak.
4. Are dynamic pruning methods benefitting from acceleration on real-world devices?

**Summary Of The Paper:**

This paper explores a static alternative pruning method for dynamic pruning methods. They propose channel attention-based learn-to-rank algorithm and  channel attention prior among all sample-specific channel saliencies. A Bayesian-based regularization is further introduced to enhance the performance.

**Summary Of The Review:**

The proposed method seems to share some novelty. The experiments are too weak. The acceleration on hardware is missing.

---

### Official Review · Reviewer_sMpW · 2022-10-24

**Confidence:** 5
**Correctness:** 2
**Technical Novelty And Significance:** 2
**Empirical Novelty And Significance:** 2
**Recommendation:** 3

**Clarity, Quality, Novelty And Reproducibility:**

The clarity and quality are good for this work.
For novelty, it is limited since the importance computation method is commonly used for pruning methods.
For reproducibility, it is also limited since hyper-parameters including learning rate, batch-size and optimizer are not provided in the paper.

**Strength And Weaknesses:**

The strengths of the proposed method are listed as below:
+ The paper is well organized and easy to follow.
+ Experiments on different network architectures are conducted.
+ Both static pruning and dynamic pruning methods are compared in experiments.

The weaknesses of the proposed method are listed as below:
- There are two key components of the method, namely, the attention computation and learn-to-rank module. For the first component, it is a common practice to compute importance using SE blocks. Therefore, the novelty of this component is limited.
- Some important SOTAs are missing and some of them as below outperform the proposed method:
(1) Ding, Xiaohan, et al. "Resrep: Lossless cnn pruning via decoupling remembering and forgetting." Proceedings of the IEEE/CVF International Conference on Computer Vision. 2021.
(2) Li, Bailin, et al. "Eagleeye: Fast sub-net evaluation for efficient neural network pruning." European conference on computer vision. Springer, Cham, 2020.
(3) Ruan, Xiaofeng, et al. "DPFPS: dynamic and progressive filter pruning for compressing convolutional neural networks from scratch." Proceedings of the AAAI Conference on Artificial Intelligence. Vol. 35. No. 3. 2021.
- Competing dynamic-pruning methods are kind of out-of-date. More recent works should be included.
- Only results on small scale datasets are provided. Results on large scale datasets including ImageNet should be included to further verify the effectiveness of the proposed method.

**Summary Of The Paper:**

This work proposes a new network pruning framework via global channel attention. In particular, it first computes the channel importance for the whole dataset via a SE block. Then 'global channel attention' is computed over the whole dataset. Then, it forces all samples to learn the same rank as the 'global channel rank'. Experiments on different network architectures and datasets are conducted to verify the effectiveness of the proposed method.

**Summary Of The Review:**

A new pruning framework is proposed in this paper. However, one of key component is not novel and key results on ImageNet are missing. More recent and high-performance SOTAs are recommended.

---

### Official Review · Reviewer_VwQA · 2022-10-25

**Confidence:** 4
**Correctness:** 2
**Technical Novelty And Significance:** 3
**Empirical Novelty And Significance:** 2
**Recommendation:** 5

**Clarity, Quality, Novelty And Reproducibility:**

Clarity
Conceptually, the ideas are clear. However, the explanation of $\phi()$ and related formulae is unclear.

Quality and Originality
A one-to-one relative comparison of dynamic vs. the proposed approach is not available. This direct comparison could serve to highlight a number of aspects of the proposed work, in terms of bringing concepts from dynamic pruning and meshing them into static approaches. In addition, the tables of results can be further updated to reflect state-of-the-art methods in the pruning domain.
While the idea of a common channel attention rank is interesting, the above issues detract from the current work.


**Strength And Weaknesses:**

Strengths
- The context and explanation provided for static and dynamic pruning are well done.

Weaknesses
- Quantitatively, on a channel to channel comparison, could the authors provide more insight in to the difference in performance and saliency between dynamic pruning approaches and GlobalPru? This could expose both the positive and negative aspects of both approaches.
- Channel level attention spans multiple ideologies and cannot be solely categorized as methods that are "local", especially over datasets, since certain methods learn inter-channel relationships over the dataset as opposed to sample-specific properties. Could the authors justify their statement in Pg. 3, Section 2.2, Lines 8-10?
- I encourage the authors to take a closer look at Figure 1 and revise it slightly so that it can be a common reference to the underlying process, especially across later sections. As constituted currently, there are certain missing elements and the flow of processes in the diagram is confusing.
- Equation 1 emphasizes objective functions which learn the mask to be applied on the weight matrices. Could the authors clarify if subsequent comparisons in the experiments section maintain this characteristic?
- The nomenclature of "prior" and "global attention during training" need to be clearly defined before being put to use. As constituted currently, they are clarified just before Section 3.3. I encourage the authors to revise the explanation in Section 3 to ensure preliminary terms are well defined before they are put to use.
- Could the authors clarify in some detail the reasoning behind the choice of expressions and formulation for the $\phi()$, and all relevant information beyond Equation 5? On first glance, there seems to be some inconsistency in notation in Equations 6 and 7.
- After establishing $\alpha, \beta$ as balancing coefficients in the main loss function, the experimental setup highlights their values to be 0.0001. Could the authors justify the choice of small values, including a comparison of the impact of varying them across a range of values?
- Results from Table 1 consistently compare against ThiNet. However, there exist a number of more advanced methods, even static pruning, which improve upon ThiNet. Could the authors provide comparisons against current works that improve upon the performance of ThiNet?
- Could the authors clarify the baseline performance across the result tables provided and whether they match the relative drop in accuracy values across baselines?
- Figure 2, the X-axis is incorrectly labelled "Interation". Please revise the label.

**Summary Of The Paper:**

The proposed work initially obtains a majority vote-based prior on the global rank of channel saliencies before forcing each sample-level channel saliency to match the global prior. In this way, the proposed work aims to use the platform of static pruning yet match the high pruning levels similar to dynamic pruning while maintaining a common channel saliency across all samples.

**Summary Of The Review:**

Justifying and addressing the points addressed in the weaknesses mentioned above could serve to highlight interesting comparisons between the proposed and existing methods.

---

### Decision · Program_Chairs · 2023-01-20

**Decision:**

Reject

**Justification For Why Not Higher Score:**

All reviewers lean toward reject and there is no author response to address reviewer concerns.

**Justification For Why Not Lower Score:**

N/A

**Metareview: Summary, Strengths And Weaknesses:**

This work develops an approach to pruning neural networks using a global ranking of channel importance.  Reviewers are unanimous in rejecting the paper and raise concerns over novelty, insufficient experimental validation, and comparison to prior work.  As there is no author response, these concerns remain unaddressed and there is no basis for accepting the paper.